# The Effect of Month and District on 100-Day In-Calf Rate in Year-Round Calving Dairy Herds

**DOI:** 10.3390/vetsci10090550

**Published:** 2023-09-02

**Authors:** Isabella S. C. Rynia, John K. House, Luke Ingenhoff

**Affiliations:** Livestock Veterinary Teaching and Research Unit, Sydney School of Veterinary Science, Faculty of Science, University of Sydney, Camden, NSW 2570, Australiajohn.house@sydney.edu.au (J.K.H.)

**Keywords:** dairy cattle, cattle reproduction, 100-day in-calf rate

## Abstract

**Simple Summary:**

The 100-day in-calf rate is a key parameter used to monitor herd reproductive performance in year-round calving dairy herds and is defined as the percentage of lactating cows that become pregnant within 100 days of calving. When 100-day in-calf rate is determined for all current lactating cows over 100 days in milk, the parameter effectively becomes a 7-month rolling average because dairy cows are typically dried off 7 months after conception. Year-round calving dairy herds typically need to achieve a 100-day in-calf rate of 40% to maintain an average calving interval of thirteen months. The objective of this study was to determine the effect of month on 100-day in-calf rate on ten dairy farms in New South Wales, Australia. The 100-day in-calf rate conformed to a seasonal fluctuation, with the lowest 100-day in-calf rates in late autumn and the highest 100-day in-calf rates in late spring. Farms located 15–140 km from the coast that experience higher maximum temperatures in summer experienced a greater fall in 100-day in-calf rate in summer and autumn compared to farms located less than 10 km from the coast. The authors propose that an assessment of herd reproductive performance against a fluctuating 100-day in-calf rate target that considers the time of year is an appropriate alternative to a constant target.

**Abstract:**

Monitoring 100-day in-calf rate (100DICR) is an integral part of the assessment of reproductive performance in year-round calving dairy herds. The objective of this study was to investigate the effect of month on 100DICR in year-round calving herds in New South Wales (NSW), Australia and determine whether a fluctuating 100DICR target is an appropriate alternative to a constant 100DICR target. The 100DICR is defined as the percentage of all current lactating cows over 100 days in milk (DIM) that conceive on or before 100 DIM. As dairy cows are typically dried off 7 months after conception, 100DICR was an approximate 7-month rolling average. Mean monthly 100DICRs were calculated with a generalised linear model for six NSW north coast herds located 15–140 km from the coast and four NSW south coast herds located less than 10 km from the coast, over a two-year period. The mean 100DICR was lowest in May at 28.62% (95%CI 28.31–28.93) and increased during winter and spring, peaking in December at 34.74% (95%CI 34.32–35.15). The observed trend was similar for north and south coast herds, although north coast herds experienced a greater change in 100DICR from the peak to a nadir of 27.58% (95%CI 27.18–27.98), a 7.15-point difference, compared to south coast herds with a nadir of 30.18% (95%CI 29.69–30.67), a 4.67-point difference between the peak and nadir. In conclusion, 100DICR is affected by month with the lowest 100DICRs observed in late autumn and the highest 100DICRs observed in late spring and early summer. Therefore, a fluctuating target 100DICR is an appropriate alternative to a constant target when assessing reproductive performance in year-round calving herds. While the district does not affect mean 100DICR per se, the district does affect the difference between peak and nadir 100DICR.

## 1. Introduction

Good reproductive performance is important for the productivity and profitability of year-round calving dairy herds and can be monitored and assessed using target-based reproductive performance parameters, such as those supported by the InCalf program [1]. The key parameters used to assess overall herd reproductive performance in year-round calving herds in the InCalf program are 100-day in-calf rate (100DICR) and 200-day not-in-calf rate, where 100DICR is defined as the percentage of cows that become pregnant within 100 days of calving [1]. The 100DICR can be viewed as a rolling average, as it is often calculated as the percentage of all cows that conceive on or before 100 days in milk (DIM) out of all lactating cows over 100DIM. Given cows are typically dried off at 7 months’ gestation, the 100DICR is usually representative of herd reproductive performance over the preceding 7-month period. The 100DICR is driven by the conception rate and 80-day submission rate (80DSR), which is defined as the percentage of cows submitted for insemination at least once within 80 days of calving [1,2]. A high 100DICR helps farms to achieve or maintain shorter calving intervals. A shorter calving interval reduces the average days in milk, improves feed efficiency, and increases the average daily milk production [1]. The InCalf program recommends a target 100DICR of 58% to achieve a 12-month calving interval [1]. However, in the University of Sydney dairy herd health program, a target 100DICR of 40% is routinely recommended. Farms that achieve a 100DICR of 40% are likely to achieve a projected average calving interval of 13 months, which is more achievable for year-round calving herds [3], whereas the 58% target is only required if a 12-month average calving interval is desired.

The 100DICR is influenced by a variety of managerial, physiological, and environmental factors including nutrition, heat detection efficiency, days to first breeding, breed, herd genetics, and climatic conditions [1]. These factors have a direct effect on the two main drivers of 100DICR. Heat detection efficiency and days to first breeding directly affect 80DSR, whereas nutrition, breed, genetics, and climatic conditions can affect the conception rate.

Changes in air temperature and humidity from season to season can affect conception rates, which in turn affects 100DICR. The optimum temperature–humidity index (THI) for fertility is <72 (or 5–25 °C and less than 50% humidity) [4,5,6]. Cavestany et al. reported up to a 20% reduction in reproductive performance associated with hotter weather in dairy cattle [7]. This seasonal reduction in fertility can result from heat stress, which in turn may cause a concurrent reduction in dry matter intake (DMI), negative energy balance, reduced oestrous behaviour, reduced quality of oocytes and pre-ovulatory follicles, increased twinning percentage, hampered endometrial function, poor-quality corpora lutea, increased early embryonic losses, and reduced conception rates [5,8]. These changes can in part be explained by the change in neuroendocrine and metabolic hormone profile of cattle experiencing heat stress [5,8,9,10,11,12,13,14,15]. The impacts of heat stress on hormone production have the greatest effect on reproductive performance from 5 weeks pre-service to 1 week post-service under consistently high THI conditions [16,17]. For these reasons, month or season has an effect on 100DICR.

The objective of this study was to investigate the effect of month on 100DICR in year-round calving herds in New South Wales (NSW), Australia and determine whether a fluctuating 100DICR target is an appropriate alternative to a constant 100DICR target when assessing reproductive performance in year-round calving herds.

## 2. Materials and Methods

### 2.1. Study Population

A retrospective study was performed utilising herd records from ten year-round calving dairy farms participating in regular reproductive veterinary visits by University of Sydney veterinarians. The four south coast farms were located in the Illawarra region of NSW and the six north coast farms were located in the Manning, Macleay, and Hunter regions of NSW. A two-year study period was included in the analysis—from January 2018 to December 2019.

The farms selected for inclusion were year-round calving herds, clients of the University of Sydney, and farms breeding predominantly by artificial insemination. The farms also needed to have complete and accurate farm records in EasyDairy (Shepparton, VIC, Australia) that included complete reproductive data for the 2018 and 2019 years. The excluded farms included those that used seasonal or batch calving patterns, farms that did not use EasyDairy, and farms that had unreliable data and majority bull-bred herds. Farms that had participated in the University of Sydney’s herd health program for less than 8 months prior to the commencement of the study period were also excluded as it was anticipated that management changes that influence 100DICR would skew the effect of month on 100DICR in these herds.

All four south coast farms had Holstein herds and maintained herd sizes of 100, 300, 350, and 575 lactating cows. Five of the north coast farms had Holstein herds and one north coast farm had a Brown Swiss herd. The north coast farms had herd sizes of 125, 200, 225, 300, 300, and 700 lactating cows. The Brown Swiss herd had a herd size of 125 lactating cows.

All ten study farms were non-housed. The cows were fed pasture and/or supplementary forage all year round with supplementary feeding of concentrates at milking time.

All ten study farms had a voluntary waiting period of 45 days. Within the two-year study period, the 80-day submission rate (80DSR) ranged from 20% to 96% and the average days to first breeding (AveDTFB) ranged from 57 to 120 days. The farm with the highest 80DSR was the 100-cow herd on the NSW south coast. The farm with the lowest 80DSR was the 225-cow herd on the NSW north coast. When farms with the highest and lowest 80DSR were excluded, the remaining eight herds had an 80DSR that ranged from 42% to 91% and an AveDTFB that ranged from 63 to 87 days.

Cows were re-inseminated if they returned to oestrus or were diagnosed non-pregnant at pregnancy testing. Decisions to re-breed individual cows were made by each farmer and based on sound commercial reasons including the number of days in milk, level of milk production, and health traits.

### 2.2. Data Collection

Herd health visits at each of the ten farms were conducted by one of two PREgCHECK (National Cattle Pregnancy Diagnosis Scheme)-accredited veterinarians. Reproductive tracts were examined with transrectal ultrasound using an Ibex^TM^ Lite portable ultrasound with a handheld L6.2 Linear Repro probe (66 mm linear array, 12 cm scan depth, 8–5 MHz; E.I Medical Imaging, CO, USA). The first pregnancy diagnosis occurred at 32–60 days on most of the study farms but increased to a maximum of 116 days on some north coast farms, where the interval between veterinary visits was up to 12 weeks.

For each of the ten farms in the study, the EasyDairy records were uploaded into the reproductive data analysis program Herdtools (JK House, University of Sydney, NSW). EasyDairy was an inclusion criterion in this study because the Herdtools program used was designed to analyse data from an EasyDairy Master.mdb file. Key EasyDairy records included cow ID, calving dates, dry-off dates, mating dates, exit dates, and pregnancy testing results. The 100DICRs were calculated by Herdtools at fortnightly intervals over the 2-year study period.

The Herdtools software calculates 100DICR on a given day as the percentage of current lactating cows over 100 days in milk that conceive on or before 100 days in milk, excluding cows intended for culling. The denominator is all current lactating cows over 100 DIM and the numerator is all current pregnant lactating cows over 100 DIM that conceive on or before 100 DIM. Given that dairy cows are typically dried off 60 days before the calving due date (or 7 months into gestation), the 100DICR calculated in Herdtools is approximately a 7-month rolling average. In the calculation of 100DICR, Herdtools identified cows intended for culling if they had a user status of “cull” or “do not breed” in EasyDairy.

The 2-year study period was the 2018 and 2019 calendar years. Each 2-year period on each farm had 53 data points, which were allocated to the month the 100DICR was calculated, irrespective of the year, so that mean 100DICRs could be calculated for each month. The 100DICR results from the 2018 and 2019 of each month were pooled to enable “Month” to consist of 12 categories for statistical analysis.

### 2.3. Statistical Analysis

Prior to determining the statistical analysis method, all 530 100DICR data points from all study farms were categorised into their nearest whole integer and a histogram was constructed. The histogram formed a bell curve shape, confirming the data were normally distributed.

Data were analysed using GenStat 18th edition (VSN International Ltd.). A generalised linear mixed model (GLMM) with normal distribution and an identity link function was conducted with month, district, breed, herd size, and year designated as the variables of interest, 100DICR as the response variable, and farm included as a random effect. Each variable was categorised. Month was categorised into the twelve calendar months, district was categorised into NSW north coast and NSW south coast, breed was categorised into Holstein and Brown Swiss, herd size was categorised into ≤225 milking cows, and ≥300 milking cows and year were categorised into 2018 and 2019.

Three generalised linear models (GLMs) with normal distribution and an identity link function were performed with month designated as the variable of interest and 100DICR as the response variable. Breed and herd size were excluded from the GLMs as factors that did not have a detectable effect on 100DICR. One GLM included all ten study farms, one GLM included only south coast farms, and one GLM included only north coast farms. The GLMs were used to attain 100DICR means and 95% confidence intervals for each month and to enable a comparison of 100DICR trends on the north coast compared to the south coast.

A chronological line chart consisting of the mean 100DICRs of all twenty-four months in the north and south coast herds was constructed to determine if the 100DICR trend was repeatable between 2018 and 2019 and to further enable a comparison of north and south coast 100DICRs.

## 3. Results

### 3.1. Generalised Linear Mixed Model (GLMM)

The GLMM determined there was a difference in the mean 100DICR of all study farms between months (*p <* 0.001), indicating that months with the highest 100DICRs were statistically different to months with the lowest 100DICRs. The mean 100DICR for 2018 was greater than the mean 100DICR in 2019 (*p* = 0.009), with back-transformed mean 100DICRs of 30.25 in 2018 and 29.26 in 2019. District (*p* = 0.75), breed (*p* = 0.48), and herd size (*p* = 0.74) did not have an effect on mean 100DICR.

### 3.2. Generalised Linear Model (GLM)—All Study Farms

The mean monthly 100DICRs and 95% confidence intervals calculated with the GLM for all study farms are presented in Table 1. The peak 100DICR occurred in December at 34.74% (95%CI 34.32–35.15) and the 95% confidence intervals show that the mean 100DICRs in October and November were not statistically different to December. The lowest mean 100DICR was in May at 28.62% (95%CI 28.31–28.93) and the 95% confidence intervals show that the mean 100DICRs in April and June were not statistically different to May. The difference between the peak 100DICR and the nadir was 6.12 percentage points.

### 3.3. Generalised Linear Model (GLM)—South Coast vs. North Coast Study Farms

The mean monthly 100DICRs and 95% confidence intervals calculated with the GLM for the south coast study farms are also presented in Table 1. The peak 100DICR occurred in December at 34.85% (95%CI 34.19–35.50) and the 95% confidence intervals show that the mean 100DICRs in September, October, November, and January were not statistically different to December. The lowest mean 100DICR was in May at 30.18% (95%CI 29.69–30.67) and the 95% confidence intervals show that the mean 100DICRs in April and June were not statistically different to May. The difference between the peak 100DICR and the nadir was 4.67 percentage points.

The mean monthly 100DICRs and 95% confidence intervals calculated with the GLM for the north coast study farms are also presented in Table 1. The peak 100DICR occurred in November at 34.73% (95%CI 34.16–35.30) and the 95% confidence intervals show that the mean 100DICRs in October and December were not statistically different to November. The lowest mean 100DICR was in May at 27.58% (95%CI 27.18–27.98) and the 95% confidence intervals show that the mean 100DICRs in February, March, April, June, and July were not statistically different to May. The difference between the peak 100DICR and the nadir was 7.15 percentage points.

The north coast farms experienced a greater change in the 100DICR between the peak and nadir compared to the south coast farms, with a lower nadir on the north coast farms. Additionally, the 100DICR on the north coast farms declined earlier than the 100DICR on the south coast farms—on the south coast farms, the mean January 100DICR was not different to the mean November and December 100DICRs, yet on the north coast farms, the mean January 100DICR, 29.31% (95%CI 28.79–29.83), was statistically significantly lower than the mean December 100DICR, 34.68% (95%CI 34.12–35.20).

The 100DICR means and 95% confidence intervals in Table 1 also show that the peak 100DICRs were not different between the north coast (a mean November 100DICR of 34.73, 95%CI 34.16–35.30) and south coast (a mean December 100DICR of 34.68, 95%CI 34.12–35.20) farms. Intra-month comparisons between the north coast and south coast farms showed that the mean 100DICRs were not different between September and December. However, the mean 100DICR was significantly lower on the north coast farms in all months from January to August. In May, the nadir north coast mean 100DICR of 27.58 (95%CI 27.18–27.98) was significantly lower than the nadir south coast mean 100DICR of 30.18 (95%CI 29.69–30.67).

Figure 1 shows a similar mean 100DICR trend in 2018 compared to 2019, particularly on the south coast farms; however, the north coast farms experienced much lower 100DICRs in the first quarter of 2018 compared to the first quarter of 2019.

## 4. Discussion

This study demonstrated a significant monthly fluctuation in the mean 100DICR on year-round calving dairy farms in NSW, with similar trends observed between districts (north coast vs. south coast). Although Figure 1 shows a similar seasonal trend in 2018 compared to 2019, the GLMM showed that the mean 100DICR in 2018 was 1% point higher than the mean 100DICR in 2019. Considering that Herdtools calculates 100DICR as the percentage of lactating cows over 100 days-in-milk that conceive on or before 100 days in milk, 100DICR conforms to an approximate 7-month rolling average. With this in mind, it is logical that the lowest mean 100DICR should be in May, as the 7-month period preceding May includes all of summer and early autumn. Similarly, it is also logical that the highest 100DICRs should be observed in November or December as the preceding 7-month period includes all of winter and spring.

The climates experienced on the north coast study farms compared to the south coast study farms are not only affected by latitude (with the two regions 250–500 km apart) but also the distance from the coast. All of the south coast study farms were located less than 10 km from the coast, whereas all of the north coast study farms were located between 15 and 140 km from the coast. These differences result in the north coast study farms experiencing higher summer temperatures and a higher THI [18]. These geographical differences explain why the 100DICRs calculated in autumn were lower in the north coast herds and why the 100DICR began to decline earlier on the north coast. On the south coast, the mean January 100DICR was similar to the November and December mean, yet on the north coast, the mean 100DICR had already started declining by January. On the north coast, the mean 100DICR in May (27.6%) was 20.6% lower than the mean 100DICR in November (34.7%). Similar results have been reported in other studies, where a reduction in reproductive performance of 20–30% has been observed after the summer period [7,19]. Comparatively, a 13.6% reduction in 100DICR was observed on the south coast in May compared to December. These differences are a likely reflection of the higher summer THIs observed on farms located further inland compared to coastal farms. Studies have postulated a THI of >72 causes heat stress, in particular a THI of >72 from 35 days before until 6 days after the day of joining reduces conception rates by up to 30% [5,20]. Peak 100DICRs are observed in late spring as the 100DICR calculation is based on the preceding 7-month period when conditions are preferable for cow fertility [21]. While the authors agree that changes in THI are likely to affect the 100DICR, other seasonal factors exist that may also contribute to changes in the 100DICR. One example is pasture quality. On the north and south coasts of NSW, dairy cows commonly graze annual or Italian ryegrass between May and November, whereas kikuyu is the most common pasture species grazed in summer and autumn. Therefore, seasonal changes affect the digestibility, energy density, and crude protein of pastures and dry matter intake in cows, which is also likely to contribute to fluctuations in cow fertility.

Management factors may exacerbate or ameliorate seasonal effects on fertility. For example, farms that use fans, cool water, sprinklers, altered milking times, and adequate shade are likely to reduce the effect of heat stress on their herd, and thus, THI alone cannot adequately predict 100DICR [5]. Further management and farm factors, including heat detection efficiency, herd size, diet management, herd genetics, breed, parity, age demographics of the herd, and herd health, may also influence the 100DICR at any time of the year [2,14,15,21,22]. This provides a possible explanation as to why there was no difference in the mean 100DICR between the two districts in the GLMM. As the ten study farms in this study were all pasture-based farms, the main options for ameliorating heat stress were the use of sprinklers over the dairy yard at milking time, the decision to milk earlier in the morning or later in the evening in summer, and the provision of high-quality forages and sodium bicarbonate in the diet.

The results from this study suggest that using a fluctuating 100DICR target to assess and monitor reproductive performance in year-round calving herds is an appropriate alternative to the conventional constant target of 40% currently used by the authors. Considering the south coast herds experienced a 4.67-point difference between the peak 100DICR and nadir 100DICR, the authors propose a fluctuating 100DICR target for coastal NSW herds less than 10 km from the coast of 37–38% in autumn, rising to a peak target of 42% in spring. Considering the north coast herds experienced a 7.15-point difference between the peak 100DICR and nadir 100DICR, the authors propose a fluctuating 100DICR target for dairy farms that experience hotter summers of 36% in autumn, rising to a compensatory peak of 43% in spring.

When assessing reproductive performance in year-round calving herds, herd veterinarians should go beyond simply comparing data against fixed targets and also consider the time of year. While a 100DICR of 36% in the spring months is considered below target in a year-round calving herd that intends to achieve a 13-month average calving interval, a 100DICR of 36% in the autumn months is not an indication of failure to achieve suitable targets. A farm that achieves a 100DICR of 36% in autumn is capable of achieving 100DICRs greater than 40% in spring.

However, it should also be acknowledged that this study lacks external validity. While the model presented in this study works well for year-round calving dairy farms in NSW, Australia, veterinarians that service dairy farmers need to be able to recognise recurring patterns that occur in their own districts. Year-round calving herds that experience higher or lower THIs than those experienced in NSW and year-round calving herds that use different farm management systems (e.g., housed herds fed total mixed rations) are likely to observe different trends compared to the trends observed by the authors. In this study, the authors used Herdtools to analyse data uploaded from EasyDairy. In Herdtools, the eligible population for calculating the 100DICR is all lactating cows over 100 DIM. As dairy cows are typically dried off 7 months after conception, 100DICRs calculated in Herdtools are effectively 7-month rolling averages. Veterinarians that use data analysis software that calculates the 100DICR differently to Herdtools will observe different trends in the 100DICR compared to those described in this study.

## 5. Conclusions

The 100DICR (when defined as the percentage of all current lactating cows over 100 days in milk that conceive on or before 100 days in milk) tends to conform to a predictable seasonal fluctuation. Therefore, the proposal of a fluctuating target 100DICR is an appropriate alternative to a constant target when assessing reproductive performance in year-round calving herds. For dairy farms that are located in other regions or use different management systems, and for veterinarians that use different data analysis programs that calculate the 100DICR differently, the trends observed may be different to the trends observed in NSW, Australia.

## Figures and Tables

**Figure 1 vetsci-10-00550-f001:**
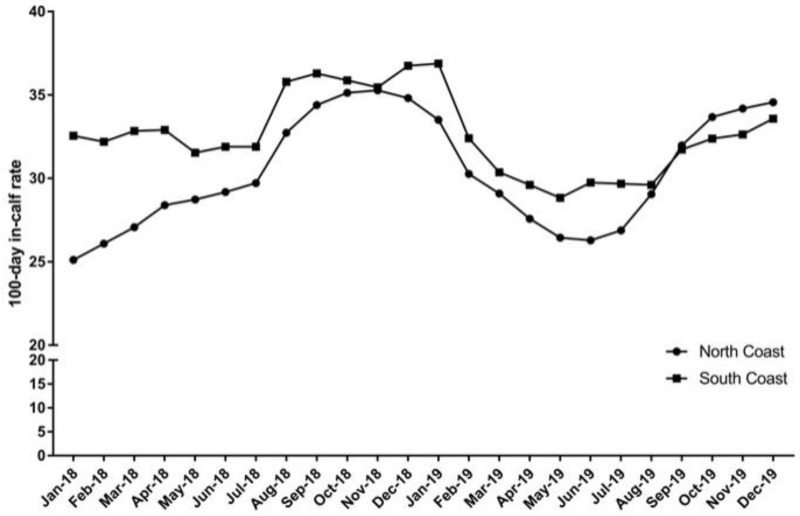
Mean 100DICR for each month of 2018 and 2019 on NSW north coast study farms and NSW south coast study farms. 100DICRs were calculated as the percentage of all current lactating cows over 100 days in milk (DIM) that conceived on or before 100 DIM. Therefore, 100DICR is an approximate 7-month rolling average.

**Table 1 vetsci-10-00550-t001:** Mean 100-day in-calf rate (100DICR) and 95% confidence intervals (95%CI) calculated by the generalised linear models (GLM) for all study farms, New South Wales (NSW) south coast study farms, and NSW north coast study farms in 2018 and 2019. The 100DICRs were calculated as the percentage of all current lactating cows over 100 days in milk (DIM) that conceived on or before 100 DIM. Therefore, 100DICR is an approximate 7-month rolling average.

	All Study Farms		South Coast Farms		North Coast Farms	
Month	Mean 100DICR (95%CI)	n	Mean 100DICR (95%CI)	n	Mean 100DICR (95%CI)	n
January	31.47 (31.07–31.87) ^cd^	60	34.72 (34.08–35.35) ^bc^	24	29.31 (28.79–29.83) ^cd^	36
February	29.82 (29.38–30.26) ^cd^	40	32.29 (31.60–32.99) ^cd^	16	28.17 (27.61–28.74) ^ad^	24
March	29.49 (29.05–29.93) ^cd^	40	31.60 (30.91–32.29) ^cd^	16	28.08 (27.51–28.65) ^ad^	24
April	29.29 (28.85–29.73) ^ad^	40	31.27 (30.56–31.95) ^ad^	16	27.98 (27.42–28.55) ^ad^	24
May	28.62 (28.31–28.93) ^ad^	40	30.18 (29.69–30.67) ^ad^	16	27.58 (27.18–27.98) ^ad^	24
June	28.97 (28.53–29.04) ^ad^	40	30.83 (30.13–31.52) ^ad^	16	27.73 (27.16–28.29) ^ad^	24
July	29.81 (29.41–30.21) ^cd^	60	32.08 (31.45–32.71) ^cd^	24	28.30 (27.78–28.81) ^ad^	36
August	31.62 (31.18–32.05) ^cd^	40	32.70 (32.01–33.39) ^cd^	16	30.89 (30.33–31.46) ^cd^	24
September	33.52 (33.08–33.95) ^cd^	40	34.01 (33.31–34.70) ^bc^	16	33.19 (32.62–33.75) ^cd^	24
October	34.30 (33.86–34.73) ^bc^	40	34.13 (33.43–34.82) ^bc^	16	34.41 (33.84–34.97) ^bc^	24
November	34.46 (34.02–34.89) ^bc^	40	34.04 (33.35–34.74) ^bc^	16	34.73 (34.16–35.30) ^bc^	24
December	34.74 (34.32–35.15) ^bc^	50	34.85 (34.19–35.50) ^bc^	20	34.68 (34.12–35.20) ^bc^	30

^a^ indicates mean 100DICR is not statistically different to the May 100DICR. ^b^ indicates mean 100DICR is not statistically different to the peak 100DICR (November for all study farms and north coast farms; December for south coast farms). ^c^ indicates mean 100DICR is statistically different to the May 100DICR. ^d^ indicates mean 100DICR is statistically different to the peak 100DICR (November for all study farms and north coast farms; December for south coast farms).

## Data Availability

Data are not publicly available but can be obtained by contacting the corresponding author.

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
