# Peer review of "The Effect of Month and District on 100-Day In-Calf Rate in Year-Round Calving Dairy Herds"

_vetsci, 2023, doi:10.3390/vetsci10090550_

Round 1

Reviewer 1 Report

The MS entitled “The effect of month and district on 100-day in-calf rate in year- 2 round calving dairy herds” written by Rynia et al analyzed the 100DICR during different month and district, however, the novelty is not clear, and the content is very simple, and I do not think it’s sufficient for the publication in Vet Sci.

General comment

The authors list the 100DICR in table and figure, however, one of them is ok for the results.

The information about calving dates, dry-off dates, mating dates, exit dates and pregnancy testing results were collected, the authors can show them also include the THI, breed, herd size for different farms in different month to enrich the content. And the authors can analyze the effect of different parities etc.

When the authors compared the differences among the months and the district, all of the differences whether significant and non-significant should be marked(a,b,ab), and the number of samples (n) should be clear.

During the same month, are there any differences between the farms located in the south coast and the north coast? The authors can compare them.

Specific comment

L8: Simple Summary can be simpler

L28: “winter and spring” should be shown in the result

L29-30: “similar in 2018 compared to 2019” were not shown in the results

L32: the conclusion should not repeat the result

Author Response

Dear reviewer 1,

Thank you for your helpful and constructive feedback.  We have revised the manuscript.  Our reply to your comments is attached in a word file.

Reviewer 2 Report

The authors investigated the effect of month and distinct on 100-day in-calf rate in year-round calving dairy herds. The topic is important/relevant because of the importance of climate conditions on reproductive performance, which is a big challenge for the dairy herd. However, despite the importance of the topic, there are some critical issues in the paper relating definition of traits, method of analysis, and interpretation of the finding that needs to be addressed before publication. My detailed concerns are listed below:

Abstract:

It may need to be updated after considering the change in M&M.

Introduction:

Very well written.

M&M:

1.      How many days after calving was the first insemination done? How was its range between herds?

2.      Were the cows re-inseminated if they were diagnosed as non-pregnant? If so, for how many times?

3.      How long after insemination the pregnancy test was done?

4.      Was the 100DICR normally distributed? If not, which method what used to normalise the data? It should be mentioned.

5.      Other factors like lactation number, milk yield, etc., may influence 100DICR. Why did you not include these parameters in the model? What about the effect of year?  

6.      Why did you do three separate analyses? You could add the State as another effect in the model.

The manuscript would be improved if the authors could include a diagram showing the reproductive cycle of studied farms indicating the calving time, first insemination, pregnancy checking, and finally 100CDIR, for an example cow.

Results:

1.      Dividing the results into three sections May not be required.

2.      Table 1 & lines 150-152: Did you have the comparisons for a few months? The way you compare the months is confusing. Do you compare all the months with each other or months within the season? Why some of the months do not have letters indicating statistical comparisons?

3.      Section 3.3. You did two separate analyses for South Coast and North Coast. How do you compare them?

Discussion & conclusion:

1.      Between-year comparison is missing in the results.

2.      Line 197-198: I assume that May included the cows who calved in Jan, inseminated in Feb, and with the assumption of successful insemination, they account for 100DICR in May. Is that correct? What is this 7-month period processing?

3.      Line 221-227: Are the studied herds different in managing heat stress?

Author Response

Dear reviewer 2,

Thankyou for your helpful and constructive feedback.  A reply to your comments is attached in a word file.

Reviewer 3 Report

This is an interesting work monitoring 100-day in-calf rate (100DICR) as an integral part of the assessment of reproductive performance in year-round calving dairy herds. Authors conclude that 100DICR is affected by month with lowest 100DICRs observed in late autumn and highest 100DICRs observed in late spring. In general, the manuscript is well written and presented. However, I have a major concern with the definition of 100DICR. Although It is clear in the introduction and discussion, the fact that is a mean of the seven preceding months should be clarified in the Simple Summary, Abstract and M&M sections. Further, it should be defined in the legends of Table 1 and Figure 1. In the present form results are confuse. For example, in the Abstract:

Lines 27 and 28 - The mean 100DICR was lowest in May at 28.62% (95%CI 28.31-28.93) and increased during winter and spring, peaking in December at 34.74%...

Lines 33 and 34 - 100DICR is affected by month with lowest 100DICRs observed in late autumn and highest 100DICRs observed in late spring…

Both sentences seem a contradiction.

Author Response

Dear reviewer 3,

Thank you for your helpful and constructive feedback.  A response to your comments is attached in a word file.

Round 2

Reviewer 1 Report

The authors have carefully corrected the paper as suggested and stated the reasons for this modification. 

Reviewer 2 Report

The authors did a great job to improve the paper.